

# Latent profiles of elite Malaysian athletes' use of psychological skills and techniques and relations with mental toughness

Vellapandian Ponnusamy[1],*, Robin L.J. Lines[2], Chun-Qing Zhang[3] and Daniel F. Gucciardi[2],*

[1] Institut Sukan Negara, National Sports Institute of Malaysia, Kuala Lumpur, Malaysia
[2] School of Physiotherapy and Exercise Science, Curtin University, Perth, WA, Australia
[3] Department of Physical Education, Hong Kong Baptist University, Hong Kong, China
* These authors contributed equally to this work.

Corresponding author
Daniel F. Gucciardi,
daniel.f.gucciardi@gmail.com

## ABSTRACT

**Background:** The majority of past work on athletes' use of psychological skills and techniques (PSTs) has adopted a variable-centered approach in which the statistical relations among study variables are averaged across a sample. However, variable-centered-analyses exclude the possibility that PSTs may be used in tandem or combined in different ways across practice and competition settings. With this empirical gap in mind, the purposes of this study were to identify the number and type of profiles of elite athletes' use of PSTs, and examine differences between these clusters in terms of their self-reported mental toughness.

**Methods:** In this cross-sectional survey study, 285 Malaysian elite athletes (170 males, 115 females) aged 15–44 years ($M = 18.89$, SD = 4.49) completed measures of various PSTs and mental toughness. Latent profile analysis was employed to determine the type and number of profiles that best represent athletes' reports of their use of PSTs in practice and competition settings, and examine differences between these classes in terms of self-reported mental toughness.

**Results:** Our results revealed three profiles (low, moderate, high use) in both practice and competition settings that were distinguished primarily according to quantitative differences in the absolute levels of reported use across most of the PSTs assessed in practice and competition settings, which in turn, were differentially related with mental toughness. Specifically, higher use of PSTs was associated with higher levels of mental toughness.

**Conclusion:** This study provides one of the first analyses of the different configurations of athletes' use of PSTs that typify unique subgroups of performers. An important next step is to examine the longitudinal (in) stability of such classes and therefore provide insight into the temporal dynamics of different configurations of athletes' use of PSTs.

## INTRODUCTION

The development of psychological skills and techniques (PSTs) alongside the physical, technical and tactical aspects of sporting performance is considered a core feature of athlete development (*Bergeron et al., 2015*). Within sporting contexts, psychological skills encompass desired personal attributes such as optimism and self-efficacy that are central to high performance (i.e., the "having" part). In contrast, psychological techniques capture the methods or processes by which individuals attain desired levels of personal attributes, such as self-talk and imagery (i.e., the "doing" part). In other words, psychological skills are developmental targets that are addressed via training in psychological techniques (*Vealey, 1988*). Evidence supports the importance of several psychological skills (e.g., confidence; *Moritz et al., 2000*; *Woodman & Hardy, 2003*) and techniques (*Brown & Fletcher, 2017*; *McCormick, Meijen & Marcora, 2015*) for high performance among athletes. Unsurprisingly, psychological skill training represents a core business for psychologists working in sport settings where high performance, innovation, and success are key (*Adler et al., 2015*; *Harmison, 2011*). Given this focus in applied practice, an important consideration for future work is understanding athletes' use of PSTs in practice and competition settings.

Much of the past work on usage of PSTs has compared successful athletes with their less successful counterparts, with a specific focus on usage patterns during training or practice (i.e., learning or applying skills or tactics in controlled environments) and within competition (i.e., when competing against other athletes). In a sample of US athletes who took part in the 2000 Olympic Games, discriminant function analysis revealed that medalists ($n = 52$) reported higher use of imagery, emotional control, and automaticity in competition than non-medalists ($n = 124$), and the skills and strategies of self-talk and emotional control distinguished these two groups of performers in terms of practice settings (*Taylor, Gould & Rolo, 2008*). Within the context of rugby union, discriminant function analysis revealed that elite Japanese players in competition settings ($n = 95$) reported higher levels of goal setting, emotional control, relaxation, and activation, and lower scores of negative thinking when compared with university level players ($n = 257$); in contrast, goal setting, imagery, and relaxation contributed most to this distinction between these two groups in terms of their usage within practice contexts (*Tanaka & Gould, 2015*). Collectively, the findings of this body of work indicate that more successful athletes use PSTs more frequently than their less successful counterparts (*Gould & Maynard, 2009*).

Despite what we have learned about athletes' use of PSTs in recent years, little is known about the different configurations of these developmental targets and methods that typify unique subgroups of performers. In particular, research in this area has been dominated by variable-centered approaches (e.g., regression and discriminant function analyses) that provide insight regarding the statistical relations among study variables averaged across a sample. An inherent assumption within variable-centered analyses where the unit of analysis is the concept or variable is that athletes' use of PSTs is homogenous and that all relations between variables generalize to the entire population

(*Bergman & Trost, 2006*). However, this assumption may be erroneous as the development and maintenance of PSTs typically incorporates both group-based and individualized components (e.g., see *Dosil, 2006*). With regard to self-talk, for example, practitioners need to consider individual differences in self-talk ability (e.g., low to high skill), cultural factors (e.g., individualistic vs collectivist cultures), and contextual dimensions (e.g., salience of different forms of self-talk) for effective training (*Van Raalte, Vincent & Brewer, 2017*). Person-centered analyses enable researchers to capture the heterogeneity of populations by identifying subgroups of individuals who share common patterns of interacting characteristics (*Bergman & Trost, 2006*). Applied to the study of athletes' use of PSTs, the potential of a person-centered approach lies in the ability to address an important yet unanswered question: are differences in athletes' use of PSTs quantitative (e.g., low, moderate, or high frequency across all attributes and methods) or qualitative (e.g., low in some techniques, high in others) in nature? In addressing this question, researchers can understand the pervasiveness of subgroups who share common patterns in their use of PSTs, and the differential relations between these profiles and hypothesized determinants and outcomes.

The second key extension offered in this study concerns the relations between unique combinations or patterns of athletes' use of PSTs with mental toughness. Drawing from recent advancements in theory and research (e.g., *Gucciardi et al., 2015*; *Hardy, Bell & Beattie, 2014*), mental toughness has been conceptualized as a psychological resource that enables individuals to attain and sustain goal-directed behavior despite varying degrees of situational demands that range from everyday stressors to major adversities (for a review, see *Gucciardi, in press*). Qualitative (*Weinberg et al., 2017*), longitudinal (*Gucciardi et al., 2015*), and experimental research (*Bell, Hardy & Beattie, 2013*) suggests that mental toughness is state-like in that it has properties that can vary or endure across situations and time, yet is open to development and enhancement. Emerging evidence provides support for the significance of mental toughness in terms of fostering high performance (e.g., *Arthur et al., 2015*; *Mahoney et al., 2014*; for a review, see *Cowden, 2017*) and maximizing adaptive associations with important psycho-social factors such as burnout (*Madigan & Nicholls, 2017*), motivation, and anxiety (*Schaefer et al., 2016*). As such, effort has been directed towards understanding the key developmental antecedents of mental toughness (for a review, see *Anthony, Gucciardi & Gordon, 2016*). Given their moderate positive effect on sport performance (*Brown & Fletcher, 2017*), it is unsurprising that psychological techniques are considered essential tools in the toolbox of mentally tough athletes (*Jaeschke, Sachs & Dieffenbach, 2016*; *Weinberg et al., 2016*) and therefore have been the focus of intervention efforts (*Fitzwater, Arthur & Hardy, 2018*; *Gucciardi, Gordon & Dimmock, 2009*). Consistent with theoretical perspectives of mental toughness as a salient resource for stressful experiences (*Gucciardi, 2017*), common to this past work are psychological techniques that enable individuals to cope with stressors and adversity (e.g., emotion regulation, arousal regulation, imagery). What remains unknown is the most effective combination of PSTs for mental toughness. Clarifying this information will have important implications for theory and practice. Substantively, it may be that one or two PSTs are most salient in terms of their contribution to the explanation of mental

toughness. If this finding is the case, an analysis of the commonalities among these dominant PSTs might shed light on the nature of mental toughness (e.g., primarily cognitive or emotion-based resource). From an applied standpoint, this information would prove fruitful in guiding practitioners' focus for psychological skill training programs.

As there are a variety of PSTs that are beneficial for athletic performance, it is important that research is directed towards identifying groups of individuals who share similar configurations or patterns of use and understanding the antecedents and outcomes of such profiles. In light of this unexplored area of research, the purposes of this study were to identify the number and type of profiles of elite athletes' use of PSTs, and examine external validity evidence of these clusters in terms of their relations with mental toughness. Given the absence of past research or theory on profiles of athletes' use of PSTs, this study was considered exploratory and therefore no hypotheses regarding the number and type of latent profiles were proposed a priori. In terms of differences between unique profiles according to athletes' use of PSTs, we expect that profiles of athletes who report using a greater number of PSTs will have higher levels of mental toughness. Which specific combination of PSTs would be most adaptive in terms of differences in self-reported mental toughness was not hypothesized a priori.

## METHODS AND MATERIALS

### Participants

In total, 285 Malaysian elite athletes took part in this study (170 males, 115 females). The term elite is used here to characterize both "semi-elite" (e.g., involved in talent development programs or selected to represent Malaysia) and "competitive elite" (e.g., internationally top tier competitive league) athletes in terms of their standard of performance, success, experience, and competiveness of sport within Malaysia and the world (*Swann, Moran & Piggott, 2015*). Athletes were recruited from national squads across a variety of team (e.g., field hockey, bola sepak) and individual (e.g., shooting, archery, taekwondo) sports. At the time of data collection, participants were aged 15–44 years ($M = 18.89$, SD $= 4.49$), had between one and 20 years of experience in their sport ($M = 4.28$, SD $= 3.38$), and completed between two and 15 training sessions per week ($M = 6.00$, SD $= 2.14$).

### Measures

We measured mental toughness using a unidimensional, eight-item tool (*Gucciardi et al., 2015*). Participants were asked to indicate how true each of the statements (e.g., "I strive for continued success" and "I am able to regulate my focus when performing tasks") are an indication of how they typically think, feel, and behave as an athlete using a seven-point response scale (1 = *false, 100% of the time* to 7 = *true, 100% of the time*). The 68-item test of performance strategies (TOPS; *Hardy et al., 2010*) was employed to measure athletes' use of several PSTs in training (goal setting, self-talk, imagery, attention control, emotional control, activation, relaxation, automaticity) and competition (goal setting, self-talk, imagery, negative thinking, emotional control, activation, relaxation,

automaticity, attentional control). Participants were asked to rate how frequently a range of situations applies to them in training and competition (e.g., "I keep my thoughts positive during competitions" and "I visualize my competition going exactly the way I want it to go") using a five-point response scale (1 = *never*, 2 = *rarely*, 3 = *sometimes*, 4 = *often*, 5 = *always*). Both tools were translated from English into Malay using forward- and back-translation procedures by an independent translator at both stages of the process (*Hambleton & Kanjee, 1995*).

## Procedures

After obtaining ethical approval from the Human Research Ethics Committee of the corresponding author's institution (HR176/2013), we contacted the high performance manager of each sport or head coach of individual teams to provide details on the aims and procedures of the study, and request permission to approach coaches and athletes. Managers and coaches who expressed an interest in participating liasied with the first author to organize a convenient time and location to distribute the survey package to the athletes in person. Consenting athletes (and their parents when aged under 18 years) completed the survey package either at the training venue prior to, or after a practice session; in situations where the time demands of a training session could not accommodate the former method, athletes took the survey home with them, completed it, and returned it at the next training session.

## Statistical analyses

Consistent with recommendations for mixture modeling (*Asparouhov & Muthén, 2013*; *Lanza, Tan & Bray, 2013*), we conducted the primary analyses in two phases. First, we used latent profile analysis (LPA) with a robust maximum likelihood estimator to determine the type and number of profiles that best represent athletes' reports of their use of PSTs during practice and competition. Aligned with an inductive approach, we first specified two profiles and sequentially increased the number of latent profiles until we arrived at the class structure that represented an optimal balance between model fit and parsimony (*Nylund, Asparouhov & Muthén, 2007*). Model comparisons were assessed using a combination of relative fix indices (Akaike Information Criteria (AIC), Bayesian Information Criteria (BIC) and its sample size adjusted version (ABIC)), ratio test (Lo–Mendell–Rubin likelihood (LMR)), and an indicator of the clarity of class allocation (entrophy). The best model is one that has the lowest value for relative fit indexes, entrophy values that are closest to 1 and larger in comparison to other class structures, and statistically significant ratio test (*Berlin, Williams & Parra, 2014*; *Nylund, Asparouhov & Muthén, 2007*). These statistical criteria were considered alongside an examination of the distinctiveness of solutions and sample size within each cluster (*Lubke & Neale, 2006*). Second, we examined age, sex (0 = female, 1 = male), and years playing experience as determinants of latent profile membership, and mental toughness as an outcome of membership within an LPA framework. The R3STEP and DU3STEP commands were used to model age, sex, and playing experience and mental toughness as an auxiliary outcome using the three-step method (*Asparouhov & Muthén, 2013*). The three-step procedure

first determines the latent classes based on the proposed indicator variables, then classifies participants, and finally relates this classification to covariates, determinants or distal outcomes. Coefficient $H$ provided an estimate of construct reliability evidence, with a value of $\geq 0.80$ considered desirable (*Hancock & Mueller, 2001*). All analyses were performed using M*plus* 7.4 (*Muthén & Muthén, 2015*) using full information maximum likelihood (FIML) to make use of all available data.

## RESULTS

Data screening procedures indicated the study variables were normally distributed in the sample (i.e., skewness/kurtosis >2), yet included three univariate outliers ($z > \pm 3.5$); the exclusion of these outliers did not alter the primary findings so they were retained for reporting purposes (*Tabachnick & Fidell, 2013*). Descriptive statistics, internal reliability estimates, and bivariate correlations among study variables are presented in Table 1. Several of the construct reliability estimates were below the recommended value of 0.80 (*Hancock & Mueller, 2001*), so it is important to keep this evidence in mind when interpreting the results of the primary analyses.

An overview of the statistical criteria employed to identify the optimal solution for practice and competition settings is detailed in Table 2. The results show that the three-class model represents the optimal solution for both practice and competition settings. First, an examination of the percentage change values in relative fit indices (AIC, BIC, ABIC) presented in Table 2 indicates there is a steep decrease in models with two to three classes at which point the degree of improvement in model fit plateaued (*Morin et al., 2011*). Second, the three-class model was the point at which the highest degree of classification accuracy was observed in competition settings (0.89), and was among the highest for practice settings (0.78). Finally, the average probability that participants were correctly classified in the given latent profile or misclassified provided additional evidence for the suitability of the three-class solution in practice and competition settings (see Table 3). Thus, we retained the three-profile solution for subsequent analyses.

An inspection of the estimated means indicated that differences between classes were primarily quantitative in nature (see Figs. 1 and 2). In terms of practice settings, class 1 ($n = 59$) reported higher levels of all PSTs than both classes 2 ($n = 72$), and 3 ($n = 154$) except for emotional control and automaticity; these latter two PSTs were comparable across all three classes. In turn, class 2 reported higher levels than class 3 on all but these two PSTs. Similar findings were observed in competition settings. With the exception of emotional control, class 1 ($n = 102$) reported higher levels of positive psychological skills and lowest levels of negative psychological skills (negative thinking) than both classes 2 ($n = 25$) and 3 ($n = 158$); similar findings were observed for the comparison of class 2 with class 3. As such, we labeled class 1 as "high adaptive use," class 2 as "moderate adaptive use" and class 3 and "low adaptive use" of PSTs.

Age, sex, and years playing experience were examined as determinants of latent profile membership (see Table 4). With regard to practice settings, older athletes and those with greater playing experience in their sport were less likely to be in the high adaptive use and low adaptive use profiles when compared with the moderate adaptive use profile.

**Table 1 Descriptive statistics, internal reliability, and correlations among study variables.**

| | 1 | 2 | 3 | 4 | 5 | 6 | 7 | 8 | 9 | 10 | 11 | 12 | 13 | 14 | 15 | 16 | 17 | 18 |
|---|---|---|---|---|---|---|---|---|---|---|---|---|---|---|---|---|---|---|
| 1 MT | (0.86) | | | | | | | | | | | | | | | | | |
| 2 Goals-p | 0.44* | (0.73) | | | | | | | | | | | | | | | | |
| 3 Imag-p | 0.33* | 0.53* | (0.68) | | | | | | | | | | | | | | | |
| 4 Attcon-p | 0.31* | 0.34* | 0.25* | (0.89) | | | | | | | | | | | | | | |
| 5 Selftalk-p | 0.36* | 0.65* | 0.52* | 0.35* | (0.71) | | | | | | | | | | | | | |
| 6 Activate-p | 0.37* | 0.44* | 0.30* | 0.38* | 0.44* | (0.65) | | | | | | | | | | | | |
| 7 Emocon-p | 0.16* | −0.05 | −0.03 | 0.39* | 0.03 | 0.24* | (0.69) | | | | | | | | | | | |
| 8 Auto-p | 0.17* | 0.12 | 0.17* | 0.00 | 0.08 | 0.10 | −0.08 | (0.72) | | | | | | | | | | |
| 9 Relax-p | 0.31* | 0.44* | 0.33* | 0.20* | 0.41* | 0.47* | 0.03 | 0.20* | (0.77) | | | | | | | | | |
| 10 Goals-c | 0.40* | 0.57* | 0.50* | 0.35* | 0.67* | 0.46* | 0.05 | 0.12* | 0.46* | (0.82) | | | | | | | | |
| 11 Selftalk-c | 0.38* | 0.62* | 0.53* | 0.42* | 0.70* | 0.46* | 0.13 | 0.05 | 0.38* | 0.65* | (0.80) | | | | | | | |
| 12 Imag-c | 0.41* | 0.57* | 0.57* | 0.38* | 0.65* | 0.50* | 0.04 | 0.16* | 0.45* | 0.62* | 0.63* | (0.69) | | | | | | |
| 13 Negthink-c | −0.24* | −0.15* | −0.07 | −0.48* | −0.24* | −0.34* | −0.38* | 0.02 | −0.05 | −0.26* | −0.31* | −0.27* | (0.78) | | | | | |
| 14 Emocon-c | 0.19* | 0.05 | 0.12* | 0.40* | 0.10 | 0.16* | 0.54* | −0.06 | 0.03 | 0.04 | 0.20* | 0.11 | −0.43* | (0.82) | | | | |
| 15 Activate-c | 0.41* | 0.52* | 0.36* | 0.33* | 0.49* | 0.66* | 0.11 | 0.25* | 0.52* | 0.54* | 0.52* | 0.58* | −0.28* | 0.12 | (0.71) | | | |
| 16 Relax-c | 0.35* | 0.52* | 0.33* | 0.34* | 0.49* | 0.53* | 0.07 | 0.18* | 0.58* | 0.53* | 0.52* | 0.53* | −0.27* | 0.03 | 0.69* | (0.79) | | |
| 17 Auto-c | 0.32* | 0.37* | 0.26* | 0.20* | 0.29* | 0.37* | 0.05 | 0.52* | 0.38* | 0.41* | 0.31* | 0.40* | −0.12 | −0.03 | 0.58* | 0.52* | (0.65) | |
| 18 Attcon-c | 0.21* | 0.31* | 0.20* | 0.32* | 0.30* | 0.41* | 0.11 | 0.01 | 0.20* | 0.40* | 0.30* | 0.38* | −0.25* | 0.01 | 0.44* | 0.40* | 0.35* | (0.69) |
| Age | 0.05 | −0.09 | 0.12 | 0.02 | 0.16 | −0.09 | 0.16* | 0.22* | 0.02 | −0.04 | 0.02 | −0.02 | −0.12 | −0.14 | 0.23* | −0.12 | −0.07 | 0.06 |
| Experience | 0.01 | 0.20 | −0.11 | −0.10 | 0.02 | 0.18* | −0.15* | −0.03 | −0.06 | 0.08 | −0.15 | 0.05 | −0.06 | 0.19* | −0.29* | 0.15 | 0.18* | 0.08 |
| Sex | 0.03 | −0.06 | −0.06 | 0.03 | 0.01 | 0.03 | 0.04 | −0.08 | 0.08 | 0.06 | −0.07 | 0.09 | 0.24* | 0.03 | −0.12 | −0.01 | 0.19* | 0.04 |
| Mean | 5.83 | 3.69 | 3.74 | 3.39 | 3.75 | 3.40 | 3.06 | 3.05 | 3.45 | 3.81 | 3.90 | 3.72 | 2.53 | 3.19 | 3.58 | 3.62 | 3.46 | 3.26 |
| SD | 0.83 | 0.64 | 0.71 | 0.55 | 0.62 | 0.54 | 0.58 | 0.61 | 0.68 | 0.65 | 0.68 | 0.61 | 0.80 | 0.74 | 0.61 | 0.69 | 0.62 | 0.42 |
| Skewness | −0.91 | −0.28 | −0.20 | 0.23 | −0.01 | 0.09 | 0.35 | −0.39 | −0.30 | −0.16 | −0.27 | −0.29 | 0.35 | 0.05 | −0.41 | −0.18 | 0.08 | −0.22 |
| Kurtosis | 0.91 | −0.10 | −0.37 | 0.21 | −0.50 | 0.29 | 0.37 | 10.09 | 0.34 | −0.14 | −0.59 | 0.15 | −0.25 | −0.14 | 0.73 | 0.40 | −0.05 | 0.21 |

**Notes:**

Readers interested in the 95% confidence intervals for these bivariate correlation can locate this information in the online supplementary material.

MT, mental toughness; Goals-p, goal setting in practice; Imag-p, imagery in practice; Attcon-p, attention control in practice; Selftalk-p, self-talk in practice; Activate-p, activation in practice; Emocon-p, emotional control in practice; Auto-p, automaticity in practice; Relax-p, relaxation in practice; Goals-c, goal setting in competition; Selftalk-c, self-talk in competition; Imag-c, imagery in competition; Negthink-c, negative thinking competition; Emocon-c, emotional control in competition; Activate-c, activation in competition; Relax-c, relaxation in competition; Auto-c, automaticity in competition; Attcon-c, attention control in competition.

* $p < 0.05$.

**Table 2 Model fit indices for all latent profile models tested.**

| | AIC | BIC | ABIC | LMR LR test $p$ value | ALMR LR test $p$ value | Entrophy |
|---|---|---|---|---|---|---|
| Practice settings | | | | | | |
| Two-Class | 3957.36 | 4048.67 | 3969.40 | 0.054 | 0.056 | 0.752 |
| Three-Class | 3833.84 (−3.22%) | 3958.03 (−2.29%) | 3850.21 (−3.09%) | 0.096 | 0.099 | 0.778 |
| Four-Class | 3810.54 (−0.61%) | 3967.60 (0.24%) | 3831.24 (−0.49%) | 0.373 | 0.379 | 0.787 |
| Five-Class | 3788.73 (−0.57%) | 3978.66 (0.28%) | 3813.76 (−0.45%) | 0.196 | 0.200 | 0.779 |
| Six-Class | 3770.64 (−0.48%) | 3993.44 (0.37%) | 3800.01 (0.36%) | 0.481 | 0.487 | 0.809 |
| Competition settings | | | | | | |
| Two-Class | 4420.40 | 4522.67 | 4433.88 | <0.001 | <0.001 | 0.858 |
| Three-Class | 4260.39 (−3.76%) | 4399.19 (−2.81%) | 4278.68 (−3.63%) | 0.01 | 0.01 | 0.886 |
| Four-Class | 4189.00 (−1.70%) | 4364.32 (−0.80%) | 4212.11 (−1.58%) | 0.142 | 0.148 | 0.841 |
| Five-Class[#] | 4148.86 (−0.97%) | 4360.71 (−0.08%) | 4176.79 (−0.85%) | 0.167 | 0.17 | 0.867 |
| Six-Class[#] | 4108.06 (−0.99%) | 4356.43 (−0.10%) | 4140.80 (−0.87%) | 1 | 1 | 0.885 |

Notes:
Number in parentheses for AIC, BIC and ABIC, change value in percentage; AIC, Akaike Information Criteria; BIC, Bayesian Information Criteria; ABIC, sample size adjusted BIC; LMR LR test, Lo–Mendell–Rubin likelihood ratio test; ALMR LR, Lo-Mendell– Rubin adjusted likelihood ratio test.
[#] Non-positive definite matrix.

**Table 3 Classification probabilities for the two to four profile solutions.**

| | Practice settings | | | | | Competition settings | | | |
|---|---|---|---|---|---|---|---|---|---|
| | 1 | 2 | 3 | 4 | | 1 | 2 | 3 | 4 |
| 1 | 0.889 | 0.111 | | | 1 | 0.964 | 0.036 | | |
| 2 | 0.045 | 0.955 | | | 2 | 0.045 | 0.955 | | |
| 1 | 0.868 | 0 | 0.132 | | 1 | 0.951 | 0.049 | 0 | |
| 2 | 0 | 0.863 | 0.137 | | 2 | 0.013 | 0.949 | 0.038 | |
| 3 | 0.043 | 0.042 | 0.915 | | 3 | 0 | 0.057 | 0.943 | |
| 1 | 0.891 | 0 | 0 | 0.108 | 1 | 0.932 | 0.011 | 0.058 | 0 |
| 2 | 0 | 0.795 | 0.090 | 0.116 | 2 | 0.085 | 0.0915 | 0 | 0 |
| 3 | 0 | 0.042 | 0.836 | 0.121 | 3 | 0.077 | 0 | 0.885 | 0.038 |
| 4 | 0.060 | 0.008 | 0.031 | 0.900 | 4 | 0 | 0 | 0.097 | 0.903 |

Note:
The five-profile and six-profile solutions are omitted here due to a non-positive definite matrix.

Within competition settings, male athletes were less likely to be in the high adaptive use profile when compared with the moderate use profile. Additionally, older athletes were more likely to be in the low used profile when compared with the moderate use profile.

We subsequently tested the degree to which these classes differed on mental toughness. With regard to practice settings, mental toughness differed as a function of class membership. High adaptive PST use athletes ($M = 6.48$, $SD = 0.38$) reported significantly ($p < 0.001$) higher levels of mental toughness than athletes classed within the moderate adaptive ($M = 5.88$, SD = 0.74, $d = 1.02$) and low adaptive PST use groups ($M = 5.21$, SD = 0.18, $d = 1.45$); the difference in mental toughness between the moderate and

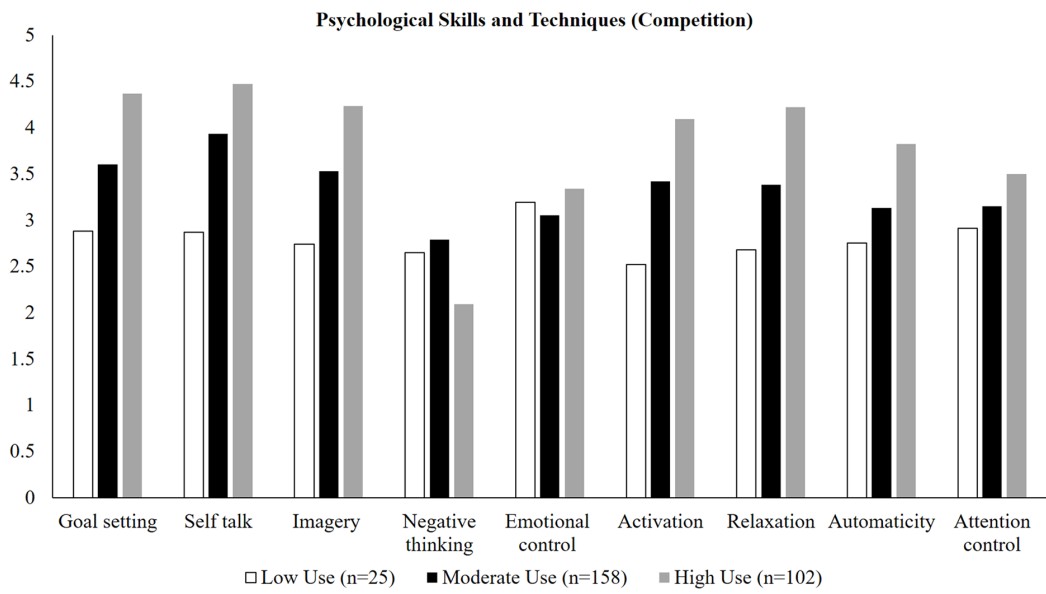

**Figure 1 Estimated means for psychological skills and techniques used in practice as a function of class membership.**

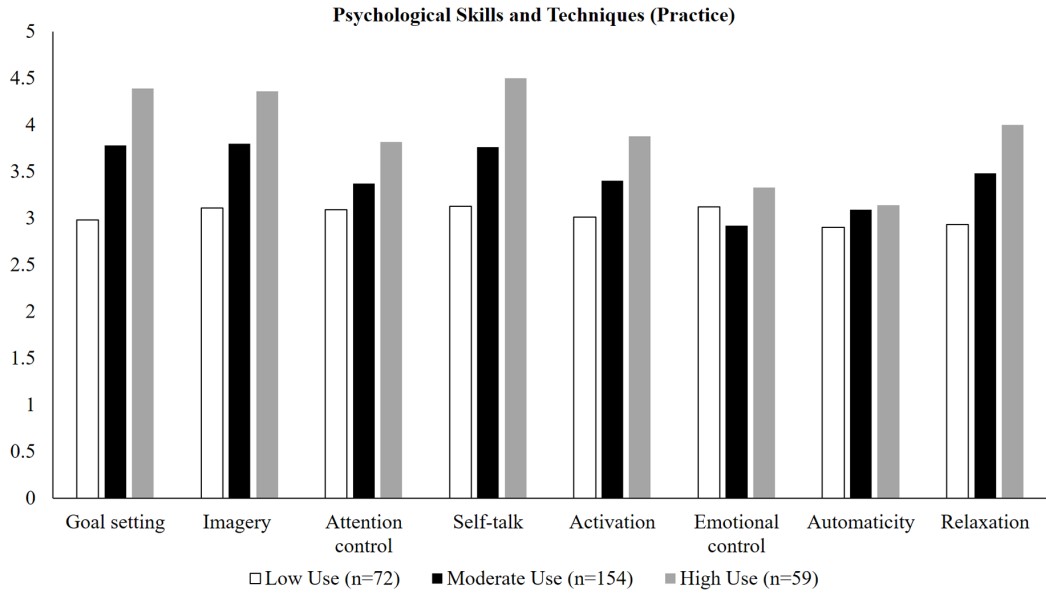

**Figure 2 Estimated means for psychological skills and techniques used in competition as a function of class membership.**

low adaptive PST use groups was also significant ($p < 0.001$, $d = 0.69$). Similar results were obtained in competition settings, whereby mental toughness differed as a function of class membership. High PST use athletes ($M = 6.36$, SD = 0.71) reported significantly ($p < 0.001$) higher levels of mental toughness than the moderate ($M = 5.61$, SD = 0.88, $d = 0.94$) and low PST use groups ($M = 5.06$, SD = 1.05, $d = 1.45$); the difference in mental toughness between the moderate and low PST use groups was also significant ($p = 0.017$, $d = 0.57$).

**Table 4 Age, sex, and years playing experience as determinants of latent profile membership.**

| Predictor | Reference class | High adaptive use | Moderate adaptive use | Low adaptive use |
|---|---|---|---|---|
| | **Practice settings** | | | |
| | High adaptive use | | | |
| Age ($n = 250$) | | | 0.15 (0.05)** | 0.04 (0.05) |
| Sex ($N = 285$) | | | 0.25 (0.40) | 0.27 (0.35) |
| Years playing experience ($n = 230$) | | | 0.22 (0.07)** | 0.11 (0.07) |
| | Moderate adaptive use | | | |
| Age ($n = 250$) | | −0.15* (0.05)** | | −0.11 (0.04)** |
| Sex ($N = 285$) | | −0.25 (0.40) | | 0.02 (0.39) |
| Years playing experience ($n = 230$) | | −0.22 (0.07)** | | −0.11 (0.05)* |
| | **Competition settings** | | | |
| | High adaptive use | | | |
| Age ($n = 250$) | | | 0.07 (0.10) | 0.15 (0.10) |
| Sex ($N = 285$) | | | 0.99 (0.50)* | 0.92 (0.50) |
| Years playing experience ($n = 230$) | | | 0.06 (0.08) | 0.11 (08) |
| | Moderate adaptive use | | | |
| Age ($n = 250$) | | −0.07 (0.10) | | 0.08 (0.03)* |
| Sex ($N = 285$) | | −0.99 (0.50)* | | −0.07 (0.29) |
| Years playing experience ($n = 230$) | | −0.06 (0.08) | | 0.05 (0.04) |

**Notes:**

Listwise deletion used for missing values on predictors. Standard errors are reported parentheses.

* $p < 0.05$.
** $p < 0.01$.
*** $p < 0.001$.

## DISCUSSION

Using a person-centered approach to investigate athletes' use of PSTs in practice and competition settings, we identified the existence of distinct profiles or subgroups of athletes who share similarities in their reported frequency. Overall, our results revealed three profiles in both practice and competition settings that were distinguished primarily according to quantitative differences in the absolute levels of reported use across most of the PSTs assessed, which in turn, were differentially related with mental toughness.

As the predominant analytical approach in past work (*Tanaka & Gould, 2015*; *Taylor, Gould & Rolo, 2008*), variable-centered-analyses exclude the possibility that PSTs may be used in tandem or combined in different ways across practice and competition settings. Our focus on uncovering different configurations of athletes' use of PSTs that typify unique subgroups of performers has shed light on a key substantive issue. Our results indicated that differences in athletes' reported use of PSTs are primarily quantitative in nature, that is, they vary in the absolute level of each developmental target and method within practice and competition settings. In terms of prevalence data, the majority of athletes were classified as moderate users of PSTs in practice (54%) and competition (55%) settings. The proportion of athletes classified as high users of PSTs was greater

in competition settings (36%) when compared with practice (21%), with the prevalence of athletes profiled as low users of PSTs was lower in competitions (9%) vs practice settings (25%). That the majority of athletes in this study reported moderate-to-high use of PSTs in practice and competition is unsurprising, given their positive effects on sport performance (*Brown & Fletcher, 2017*; *McCormick, Meijen & Marcora, 2015*) and the elite status of the sample. Across both practice and competition settings, the largest differences between classes in absolute use scores were observed for goal setting, self-talk, imagery, activation, and relaxation. Substantively, these PSTs are primarily cognitive (e.g., imagining alternative performance strategies) and arousal-focused (e.g., psyching oneself up/down for competition) in nature. An important next step is to examine the longitudinal (in)stability of such classes and therefore provide insight into the temporal dynamics of different configurations of athletes' use of PSTs.

We assessed external validity evidence of these classes of differing use of PSTs in terms of their relations with mental toughness. Only one study to date has examined the associations between mental toughness and athletes' use of several PSTs. Correlational analyses of a sample of 107 university or county level athletes revealed positive associations between mental toughness and self-talk, emotional control, goal setting, relaxation, and activation ($0.24 < r > 0.37$), and an inverse relation with negative thinking ($-0.47$) within competition (*Crust & Azadi, 2010*). For practice settings, total mental toughness was associated with automaticity, emotional control, relaxation, and self-talk ($0.24 < r > 0.35$). In contrast, we observed primarily moderate associations between mental toughness and athletes' use of all PSTs in practice and competition settings (see Table 1). One explanation for this discrepancy is that we only sampled elite athletes who have greater access to psychological servicing through the national institute of sport. The bivariate associations between mental toughness and PST use observed in the current study are broadly consistent with qualitative work in which scholars have emphasized the importance of teaching athletes techniques to cope with stress and adversity (e.g., *Jaeschke, Sachs & Dieffenbach, 2016*; *Powell & Myers, 2017*; *Weinberg et al., 2016*), most of which are captured in the TOPS. A key consideration for PST is the simulation of stressors and adversities to provide athletes with opportunities to test out different techniques (*Weinberg, Freysinger & Mellano, 2018*). Our findings provide indirect support for this assertion in that the associations between mental toughness and PST use were similar across practice and competition settings.

With regard to the results of the person-centered analyses, there were salient distinctions in mean levels of mental toughness between the three classes that confirmed the quantitative nature of the differences in the configurations of athletes' use of PSTs. Specifically, the high PST use class reported the highest levels of mental toughness; the low PST use class reported the lowest levels of mental toughness; and the moderate PST use class reported mental toughness levels between these two classes. Broadly, these findings are consistent with past intervention work in which athletes who have received psychological skills training packages including multiple techniques have evidenced increases in their mental toughness (*Fitzwater, Arthur & Hardy, 2018*; *Gucciardi, Gordon & Dimmock, 2009*). An examination of the variability in mental toughness scores provides further support for this substantive

finding, such that the least amount of variation was observed in the high PST use class and the most variation in the low use class. Collectively, these findings shed light on the nature of mental toughness and suggest that it may best be conceptualized as a resource that is characterized by cognitive and motivational factors. The broad use of multiple PSTs for athletes with high levels of mental toughness suggests that they have at their disposal a greater repertoire of techniques from which to draw to manage the various stressors and adversities within their performance environment. Equally, it could also be that greater PST use is an outcome of high mental toughness. Determining the causal characteristics between mental toughness and PSTs remains an important avenue for future research.

Key strengths of this study include a modest sample of elite athletes, direct test of the hypothesis that PSTs may be used in tandem or combined in different ways across practice and competition settings, and incorporation of error in profile classification for the examination of differences in mental toughness between the latent classes. Nevertheless, there are several limitations of the current study that might motivate future research. First, a limitation of the TOPS is that it measures the frequency of athletes' use of PSTs only and therefore excludes information regarding the effectiveness of their implementation or utilization. As the majority of PSTs assessed with the TOPS are primarily cognitive and arousal-based in nature, it is also important to consider additional measures that capture emotionally-salient PSTs. Second, our focus on Malaysian elite athletes means it is important that future research examines the extent to which these findings generalize to other cohorts of athletes (e.g., culture), and test the invariance of latent profiles across different subgroups (e.g., sex, sport type; *Olivera-Aguilar & Rikoon, 2018*). Third, we relied on one outcome variable to assess external validity evidence of the latent profiles. Future research could expand on our efforts to focus on a variety of antecedent (e.g., contextual, social or organizational factors) and outcome (e.g., decision-making, perceptions of stress) variables, which could include both subjective assessments from the self or informants (e.g., coaches) and objective performance data.

## CONCLUSION

The burgeoning literature on PSTs among elite athletes has been limited by the dominance of variable-centered analyses that do not account for the unique ways in which athletes may use multiple skills or methods in conjunction with each other. By adopting an alternative, person-centered approach, this study offers several contributions to theoretical and applied perspectives on athletes' use of PSTs and their relations with mental toughness, namely, the use of PSTs is primarily quantitative in nature, and that classes with more frequent use report higher levels of mental toughness.

### Funding

Data collection in Malaysia was supported by a grant from Institut Sukan Negara Malaysia (ISNRG 01-2014-07-2014). The funders had no role in study design, data collection and analysis, decision to publish, or preparation of the manuscript.

## Grant Disclosures

The following grant information was disclosed by the authors:

Institut Sukan Negara Malaysia: ISNRG 01-2014-07-2014.

## Competing Interests

The authors declare that they have no competing interests.

## Author Contributions

- Vellapandian Ponnusamy conceived and designed the experiments, performed the experiments, authored or reviewed drafts of the paper, approved the final draft.
- Robin L.J. Lines conceived and designed the experiments, performed the experiments, authored or reviewed drafts of the paper, approved the final draft.
- Chun-Qing Zhang conceived and designed the experiments, authored or reviewed drafts of the paper, approved the final draft.
- Daniel F. Gucciardi conceived and designed the experiments, performed the experiments, analyzed the data, prepared figures and/or tables, authored or reviewed drafts of the paper, approved the final draft.

## Human Ethics

The following information was supplied relating to ethical approvals (i.e., approving body and any reference numbers):

Ethical approval was obtained from the Human Research Ethics Committee of Curtin University.

## Data Availability

Open Science Framework: https://osf.io/5adf8/?view_only=afd0998c2d40415 dbd7d76a7e2ebbf1f.

## Supplemental Information

Supplemental information for this article can be found online at http://dx.doi.org/10.7717/peerj.4778#supplemental-information.

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
