# Peer review of "Latent profiles of elite Malaysian athletes’ use of psychological skills and techniques and relations with mental toughness"

_PeerJ, doi:10.7717/peerj.4778_

## Round 0.1 · original submission · Major Revisions

Two reviewers with substantial expertise have examined your submission and provided critiques for consideration in the review process. The reviewers express interest in your submission on various accounts and offer encouragement for this line of inquiry. I agree with the positive sentiment expressed in the reviews. Reservations are, nonetheless, evident in the reviewers' critiques. Their observations are presented with clarity so I'll not risk confusing matters by belaboring or reiterating their comments. While I might quibble with the occasional point, I note that I regard the reviewers' opinions as substantive and well-informed. I believe that all of the highlighted reservations require contemplation and appropriate attention in revising the document if it is to contribute appropriately to Peer J and the extant literature.

After reflecting upon my reading of the manuscript and the reviewers' observations, I have concluded that your submission is presently not suitable for publication in Peer J. Although I share the reviewers’ interest in your report, I also share their reservations. I have, however, decided to provide you with the opportunity to use their observations to good effect by inviting you to revise and resubmit the manuscript before I come to a final decision on its disposition at Peer J. If you decide to take advantage of this opportunity by resubmitting the manuscript, I will provide an unambiguous decision after evaluation of the revised report.

I ask that all reviewer observations be addressed in revising the manuscript. How and where the reviewers’observations are addressed (or rebutted) should be explained on a point-by-point basis. Some matters will require little more than minor editing. Other matters are likely to require much more substantial contemplation and effort. I look forward to reading your revised submission and the accompanying line-by-line responses. I anticipate that both will make for interesting reading.

I note that I would reject your manuscript at this point if I was certain that the difficulties identified by the reviewers were impossible to overcome. Please understand, however, that this opportunity to revise and resubmit the manuscript does not guarantee that it will ultimately be accepted for publication. The revised manuscript and your line-by-line responses will be sent out for further reviewer input to ensure that my decision is appropriately informed.

Tsung-Min Hung, Ph.D.
PeerJ editor
Distinguished professor,
Department of Physical Education,
National Taiwan Normal University

Reviewer 1 ·

Basic reporting

There are some theoretical backgrounds are not so clear as follow.

First, I am confused by authors’ definitions of psychological skills and techniques on line 58-61. The authors’ defined psychological skills as “...encompass developmental targets or desired attributes that are central to high performance...” I don’t understand what this means. According to Oxford dictionary, skill is defined as “the ability to do something well.” The Collins Dictionary, skill refers to “... a type of work or activity which requires special training and knowledge...”

As to technique you defined as “... capture the methods or processes by which individuals attain and sustain such outcomes (e.g., self-talk, imagery). Again, the Oxford dictionary defined it as “A way of carrying out a particular task, especially the execution or performance of an artistic work or a scientific procedure.” The Collins Dictionary defined technique as “... a particular method of doing an activity, usually a method that involves practical skills...”

By your definition of skills and techniques, I don’t understand what is (are) your research questions. Also, from your methods and materials section, you used “TOPS” to measures athletes PSTs in training (i.e., goal setting, self-talk, imagery, attention control, emotional control, activation, relaxation, automaticity) and competition (goal setting, self-talk, imagery, negative thinking, emotional control.) Are these psychological skills or techniques?

Second, you have a false citation. On line 62-63, you cited Van Yperen and colleagues (2014) support the importance of psychological skills on high performance. How come this is true because Van Yperen et al., 2014 talked about achievement goals.

Third, you said that mental toughness is “state-like in that it has properties that can vary or endure across situations and time...” If this is true, it seems contradictory against Jones’ definition (2002) that mental toughness is “the natural or developed psychological edge...” Also, by your methods and materials section, you used Gucciardi and colleagues (2015) measure to assess participants’ mental toughness. The items you demonstrated (e.g., I strive for continued success and I am able to regulate my focus when performing tasks) are trait or disposition.

Fourth, the link between PST and mental toughness needs to be a further explanation on line 116-117. You simply stated that “...one or two PST are most salient to mental toughness...” By what? Why? Please explain more.

Experimental design

1. You said you sampled 285 Malaysian elite athletes participated in your study. How elite are they? What are their elite records? Please explain more.

2. Your participant's age ranged from 15-44 years. How come 44 years are still engaged in competitive sports. How many percentages they are represented in the whole sample. Please show the describe it.

3. Before doing your main statistical analyses, did you find any outliers, skewness, the kurtosis of your study? What was the normality of your data?

4. I suggest you use traditional format to present your methods and material, such as participants, measurements, procedures, statistical analyses...

Validity of the findings

The general findings are significant and meaningful. But I found on line 213 you mentioned: “...high adaptive PSTs use athletes reported significantly higher levels of mental toughness...” What means to high adaptive PSTs stands for? Please explain it.

Additional comments

The paper is thoughtful but needs lots of revisions to make this paper acceptable.

Reviewer 2 ·

Basic reporting

Comments provided in text.

Experimental design

Comments provided in text.

Validity of the findings

Comments provided in text.

Additional comments

This article used LPA to identify underlying classes of athletes who vary in their use of PSTs. These profiles were then used to predict mental toughness. The manuscript is well written and tackles a novel area of the literature using an analytical technique that is quite popular at the moment. I do, however, have some reservations about the paper in its current format and have made comments/suggestions for consideration.

Introduction
1. Lines 71 to 73: Could the authors clarify what they mean by “whereas the skills and strategies of self-talk and emotional control distinguished these two groups of performers in terms of practice settings (Taylor, Gould, & Rolo, 2008).”
2. Lines 76 to 78: Similar to the above, it’s not clear what is meant by “in contrast, goal setting, imagery and relaxation contributed most to these distinction between these two groups in practice contexts (Tanaka & Gould, 2015).” Details on the above would help to support the statement in lines 79 to 80 suggesting that athletes who are more successful tend to use PSTs more frequently.
3. Line 77 to 78: Consider rephrasing the sentence.
4. Lines 88 to 90: It would be useful to see this statement expanded upon “However, this assumption may be erroneous as the development and maintenance of PSTs typically incorporates both group-based and individualized components (for examples, see Dosil, 2006).” Can the authors elaborate on an example of what they’re referring to?
5. Lines 100 to 114: More detail needs to be provided about mental toughness, including the importance of the construct for sport (e.g., performance), perspectives on dimensionality, major developments, etc. There have been several reviews published in 2017 that the authors could draw on in this regard.
6. Some detail on the types of psychological skills and techniques associated with mental toughness would be useful in the section on PSTs and mental toughness.
7. Lines 126 to 129: I disagree that omitting hypotheses is the best approach. Based on the literature you present and what we know about mental toughness, surely an argument could be made about the types of PSTs you would expect mental toughness to correspond with?

Method
1. Lines 137 to 151: Evidence on the reliability and validity of the measures that were used needs to be included, especially with respect to cross-cultural applicability.
2. It’s difficult to see how a person with a year of experience in their sport could be part of a “national squad.” While this may be true, the authors should elaborate on the criteria that athletes needed to meet in order to be included on these national squads.
3. What about parental consent for participants that are not legally considered adults? Was this part of the ethical clearance submission (i.e., did the authors get ethical clearance to collect data from minors?)? Did parents’ consent to their children participating in the study (based on the IC document attached, it doesn't seem that parental consent was obtained)?
4. In the informed consent document that outlines the measures participants completed, there is no reference to the TOPS in the alphabetized list. Did the authors obtain clearance to administer this inventory?

Results
1. Lines 192 to 193: What specifically are the authors referring to when they state “…. the degree of improvement plateaued (Morin et al., 2011)?” This is quite a subjective statement and requires further clarification (perhaps more detail would be useful).
2. Lines 196 to 198: The authors make the following statement: “Finally, the average probability that participants were correctly classified in the given latent profile or misclassified provided additional evidence for the superiority of the 3-class solution in practice (>.85) and competition settings (>.92),” but this is only interpretable alongside the average probabilities associated with other class solutions. I suggest including these in the table along with the model fit indices.
3. It would be useful to have confidence intervals reported for all zero-order correlations.
4. Some of the internal consistency estimates are low (< .70). Could the authors elaborate on how they determined the internal consistency of these scales to be adequate, particularly in light of the fact that information about the reliability and validity of the instruments is not reported in the method section?
5. Why did the authors not test for potential confounds (such as age, playing experience, sex) and control for them where relevant? This is particularly relevant considering the extensive age range and length of participation range quantity discrepancy in the current sample, which seems an important issue to elaborate on when reporting the results.

Discussion
1. The authors report only one study has looked at mental toughness and PSTs. While this may be accurate, the discussion could elaborate on broader research that has indirectly found mental toughness is related to various types of PSTs.
2. Lines 269 to 271: This statement is quite speculative; the cross-sectional nature of the data cannot definitively imply PSTs as precursors of mental toughness (they could be outcomes too). Some revision to the phrasing here seems necessary.

---

## Round 0.2 · Major Revisions

I now have received two re-reviews. Although both reviewers were general positive to your revision, one reviewer has pointed out some addition revisions are required to improve the quality of this manuscript. Their observations are presented with clarity so I'll not risk confusing matters by belaboring or reiterating their comments. I believe that all of the highlighted reservations require contemplation and appropriate attention in revising the document if it is to contribute appropriately to Peerj and the extant literature. Please revise or refute according to the reviewer's comments and provide a point by point reply in addition to the revised manuscript.

Tsung-Min Hung

Reviewer 1 ·

Basic reporting

The revised version has improved much comparing to the first draft. Especially, I concerned with the rationale about the skills and techniques, as well as the association between psychological skills and mental toughness. The revised version addressed these very well.

Experimental design

Research questions are clearly stated, and the methods are reasonable and acceptable.

Validity of the findings

The specific contribution of this study was to adopt a person-centered approach to analyze the different profiles of PST used in training and competition, and their differences in metal toughness. Results not only found different profiles of PST used in training and competition but also found high level of PST use was associated with higher level of mental toughness. Results bring significant contributions to the field of sport psychology. Also, the research approach is inspiring for future researchers.

Additional comments

Congratulations to successfully accomplished a significant paper that address the relationship between PST and mental toughness. Results were inspiring.

Reviewer 2 ·

Basic reporting

No comment

Experimental design

No comment

Validity of the findings

No comment

Additional comments

Overall, I thought the authors did an excellent job with the revisions. It was great to read the revised version. That said, I still have a few reservations that I think need to be addressed before publication. My comments are below.

Introduction

Lines 116-120: It’s noted the authors have attempted to address the issue surrounding mental toughness and performance. The references provided, however, could better reflect the breadth of literature available rather than focusing on two major groups of researchers alone. There are at least two reviews on mental toughness and achievement/performance, one in sport and one in more general contexts. A review along these lines is certainly preferred over single papers that address the same issue. The two articles I refer to are below; at least one should be mentioned in light of the statement on mental toughness and performance.

Cowden, R. (2017). Mental toughness and success in sport: A review and prospect. The
Open Sports Science Journal, 10. doi:10.2174/1875399X01710010001

Lin, Y., Mutz, J., Clough, P., & Papageorgiou, K. (2017). Mental toughness and individual differences in learning, educational and work performance, psychological well-being, and personality: A systematic review. Frontiers in Psychology, 8, doi:10.3389/fpsyg.2017.01345

Results

Lines 213 to 215: A reference is required for statements about acceptable levels of normality and outlier detection cut-offs.

Lines 241 to 247: The additional analyses predicting class membership are interesting and as one might have expected. This doesn’t necessarily negate the issue of needing to control for at least some of these variables in the analysis involving mental toughness (the most important outcome variable). How do we know that the profiles show differences in mental toughness when you control for relevant variables such as playing experience? I didn’t pick up in the text where the authors had reported controlling for some of these demographics in this analysis. At a very minimum, the authors should check and report the relationships between mental toughness, age, sex, and playing experience. If none are associated with mental toughness, one could see an argument being made for omitting them from the analysis involving mental toughness. Without reporting these the reader is left wondering what effect the large age and playing experience ranges, in particular, have on the outcome of the analysis with mental toughness.

---

## Round 0.3 · Minor Revisions

I have read through your point to point reply to reviewers' comment and your tracked change file before deciding whether sending out to reviewers. However, I couldn't find your action taken to reply to the first and second comment made by reviewer 2. Please double check your revision and resubmit so that I can complete the evaluation and make a corresponding decision.

Tsung-Min Hung, PhD., FNAK
PeerJ editor
Research chair professor,
Department of Physical Education,
National Taiwan Normal University

---

## Round 0.4 · accepted · Accept

I have read through your reply to the reviewer's comment and your revised manuscript. I am satisfied with your response and decided that there is no need to send to the reviewer. You and your coauthors have my congratulations. Thank you for choosing PeerJ as a venue for publishing your research work and I look forward to receiving more of your work in the future.

Tsung-Min Hung, PhD., FNAK
PeerJ editor
Research chair professor,
Department of Physical Education,
National Taiwan Normal University

#